

# Targeting the soil quality and soil health concepts when aiming for the United Nations Sustainable Development Goals and the EU Green Deal

Antonello Bonfante[1], Angelo Basile[1], Johan Bouma[2]

[1]Institute for Mediterranean Agricultural and Forest Systems - CNR-ISAFOM, Ercolano, 80056, Italy

[2]Em. Prof Soil Science, Wageningen University, the Netherlands

*Correspondence to*: Antonello Bonfante (antonello.bonfante@cnr.it)

**Abstract.** The soil quality and soil health concepts are widely used as soils receive more attention in the worldwide policy arena. So far, however, the distinction between the two concepts is unclear and operational procedures for measurement are still being developed. A proposal is made to focus soil health on actual soil conditions, as determined by a limited set of indicators that reflect favourable rooting conditions. In addition, soil quality can express inherent soil conditions in a given soil type (genoform) reflecting the effects of past and present soil management (expressed by various phenoforms). Soils contribute to ecosystem services that, in turn, contribute to the UN Sustainable Development Goals and, more recently, to the EU Green Deal. Relevant soil ecosystem services are biomass production (SDG2: zero hunger), providing clean water (SDG6); climate mitigation by carbon capture and reduction of greenhouse gas emissions (SDG13: climate action) and biodiversity preservation (SDG15: life on land). The use of simulation models for the soil-water-atmosphere-plant system is proposed as a quantitative and reproducible procedure to derive single values for soil health and soil quality for current and future climate conditions. Crop production parameters from the international: "yield-gap" program are used combined with soil-specific parameters expressing the effects of phenoforms. These procedures focus on the ecosystem service: biomass production Other ecosystem services are determined by soil-specific management to be based on experiences obtained in similar soils elsewhere or by new research. A case study, covering three Italian soil series, illustrates the application of the proposed concepts, showing that soil types (soil series) acted significantly different to effects of management also in their reaction to climate change.

## 1 Introduction

The soil receives increasing attention in the research and policy arena focusing on its capability to perform a number of functions. The concepts of soil quality and soil health are often used to express this capability, but this is only meaningful when these two concepts are clearly defined and can be established with operational and reproducible methods. So far, this methodology has not been developed. Moreover, methods to assess soil health and soil quality derive their significance from societal relevance in a broad ecosystem context as defined by the United Nations in 2015, in terms of seventeen Sustainable Development Goals (https://www.un.org/sustainabledevelopment-goals) and by the 2019 Green Deal of the European Union



(https://ec.europa.eu/info/strategy/european-green-deal). In the United States, Soil health is supported by the policy arena and
is being studied by at least three Institutes: Cornell University, The National Soil Health Institute and the US Dept. of
Agriculture. The new research and innovation program of the European Union for the period 2021-2027, "Horizon Europa"
has defined five MISSION areas, among them: "Soil Health and Food", recognizing the importance of soils for sustainable
development. Soils are now clearly on the international research agenda!
To allow operational use of the soil health concept, a clear measurement methodology is needed. So far, Cornell University
has proposed a method to measure soil health, defining a set of indicators and a procedure resulting in a number between 1
and 100 ranging from highly unhealthy to shiningly healthy. This procedure will be discussed in this paper. The term soil
health is attractive not only because of its analogy with human health that facilitates communication with the public but also,
and particularly, because soils are biologically active as are humans. The older term soil quality that has been used for decades
(e.g., Bünemann et al. 2018) has a more sterile character that could also apply to, e.g., nuts and bolts. According to some (e.g.,
USDA, 2019), soil health and soil quality have the same meaning. This, however, is not logical because why introduce a new
term when it has the same meaning as the old one? The objective of this article is to propose that both terms can be distinguished
allowing a useful distinction between actual versus inherent conditions. The proposed concepts have been illustrated in an
Italian case study.

## 1.1 The soil quality concept

Soil quality has been defined as: *"the capacity of a soil to function within ecosystem and land-use boundaries to sustain*
*biological productivity, maintain environmental quality and promote plant and animal health"* as quoted by Bünemann et al.
(2018) in a comprehensive review of more than 250 scientific papers covering soil quality. The authors conclude that, in
contrast to the quality of water, air, and nature, there still is no universally accepted method to measure soil quality. This is a
serious problem, limiting application in practice and in environmental rules and regulations.

## 1.2 The soil health concept

Soil health has been defined in the US as *"the continued capacity of the soil to function as a vital living ecosystem that sustains*
*plants, animals and humans"*. Indicators for soil health have been defined in the USA: 19 by Cornell University (Moebius-
Clune et al., 2017), 31 by the National Soil Health Institute (http://soilhealthinstitute.org) and 11 by the US Department of
Agriculture (USDA, 2019). How these indicators are combined into a single soil health parameter for a given soil is presented
by the Cornell protocol. Only three texture classes of soils are distinguished: coarse, medium and fine. For each texture class,
measurements for each indicator are assembled for soils at different locations in that particular texture class and a frequency
curve of values is constructed. Obviously, such curves become more diagnostic as more data become available. When placed
on the frequency curve, any new observation of the indicator will obtain a number between 0 and 100. This procedure is
repeated for every indicator and in the end all numbers will be averaged producing one characteristic number for soil health
for that particular soil, which is quite attractive for communication purposes. The frequency curve also allows the distinction





of a threshold frequency value above which the particular indicator exceeds a critical environmental threshold value, sometimes
defined by environmental laws and regulations. In their reporting red, orange, yellow and green colours are used to indicate
whether or not this occurs. A red label indicates that a given threshold is exceeded and that action is needed, possibly to be
based on favourable management experiences obtained elsewhere in soils of the same texture class or by new research. This
is attractive because it can directly result in management advice. In an example presented by Moebius-Clune et al. (2017) on
page 73, values for twelve indicators are presented, three of which with a red label: "surface hardness", "aggregate stability"
and "active carbon content", suggesting a need for corrective measures. But what does this imply for soil health? A soil is
unhealthy if only one or more indicators are red? And how to interpret an average value for all twelve quite different indicators
with different colours?
Also, a question can be raised about the large number of indicators for soil health in the three US systems. Why not primarily
consider demands by roots as they link plants with the soil? A number of conditions do not allow root growth: e.g., presence
of excessive amounts of chemical pollutants, salty soils (solonchack), alkaline soils (solonetz) and very acid soils with low pH
values. Soils with such properties are clearly unhealthy. Otherwise, roots require: (i) temperatures that allow growth; (ii) soil
structure that allows easy accessibility of the entire soil volume, allowing roots to reach their genetically determined depth;
(iii) adequate water, air and nutrient availability during the growing season; (iv) adequate infiltration rates of water at the soil
surface; and (v) adequate organic matter content and the associated biological activity that is essential for many soil functions,
including nutrient uptake by plants. These five parameters can be measured at a given time and place and the reports by
Moebius Clune, (2017) and USDA (2019) contain detailed descriptions of measurement methods.
Parameters to be measured at a given point in time should have a semi-permanent character to be diagnostic. Temperature and
nutrient status are quite variable, the latter high at the moment of fertilization and increasingly lower as the crop adsorbs
nutrients. Of course, this is different in nature areas where inherent nutrient contents are important to allow particular types of
vegetation to develop. However, nutrient deficiencies in agricultural soils can be rapidly corrected by fertilization and the
nutrient status, though essential for root growth, is therefore less suitable as a parameter in agricultural soils. Soil structure,
excluding a limited period after soil tillage, is more permanent and governs infiltration rates and soil water and air regimes as
a function of weather conditions and groundwater dynamics. Soil structure is therefore suitable as a parameter. Aggregate
stability is a measure for soil resistance to deformation but the method has been criticised as being unrepresentative (e.g.,
Baveye, 2020). The use of penetrometers may be more effective to measure mechanical resistance affecting root penetration.
Biological activity is subject to an even longer time span than compaction: increasing the organic matter content of soils may
take several years. The organic matter content is, therefore, a suitable parameter and many measurement methods are available,
including rapid methods applying proximal sensors. More detailed measurements of biodiversity have been defined by
Moebius-Clune, (2017) and for the LUCAS soil database (Orgiazzi et al., 2018), requiring laboratory measurements.
In conclusion, parameters for soil health for a given soil type at a given time and place, are: (i) soil structure, expressed by
descriptions in soil survey reports and supported by bulk density values and measured infiltration rates, and, possibly, by
penetrometer values, (ii) water and air regimes, as estimated by drainage class in soil survey reports, can be expressed indirectly





by the widely used but static parameter: "available water" defining the water content between two pressure heads, which,
however, poorly represent natural dynamic soil water and air regimes. Dynamic modeling presents more realistic data as will
be discussed later (e.g., Bouma, 2018, Bonfante et al., 2019) and (iii) organic matter contents.
Nevertheless, the procedure based on the three parameters mentioned above produces three separate values. Back, therefore,
to the definition of soil health that mentions "functioning of soils", whereby soil contributions to biomass production is a key
function, among six other defined functions (EC, 2006). The degree by which biomass production is affected by the three
separate parameters remains unclear. An integrated approach is therefore needed and can be obtained by simulating the soil-
water-atmosphere-plant system.

**1.3 Still a role for soil quality?**

The soil health concept offers one fundamental problem. A sandy soil and a clay soil can both be healthy, but they obviously
have quite different water and nutrient regimes and use- potentials. But differences among soils can be expressed by the soil
quality concept when considering inherent properties of soils as expressed in soil classification, like texture, which is most
stable among all soil parameters (see also Moebius-Clune, 2017). In analogy with human health, soil health for a given soil at
a given time expresses the actual condition expressed by the parameters discussed above, just like a doctor assesses the health
of a patient at a given time applying a set of tests. As discussed, different health values can be found in the same soil type as a
function of past management, such as compaction, soil crusting followed by runoff, erosion, etc., as illustrated in the Italian
case study presented below. However, the range of such soil health values is characteristically different for every soil type and
can, therefore, function as a measure of soil quality. Droogers and Bouma (1997) have distinguished genoforms, expressing a
given soil classification, but also phenoforms of that particular genoform, as a function of different forms of management with
strong effects on soil functioning (e.g., erosion, compaction, crust formation). Traditional soil survey interpretations are based
on so-called "representative profiles" for each mapping unit on the soil map, based on permanent Taxonomic soil criteria,
correctly ignoring in the classification context the effects of management which would lead to highly variable classifications.
Different phenoforms of a given genoform can, however, function quite differently and this cannot be ignored when
considering soil health.

**1.4 Simulating the soil-water-atmosphere-plant system to obtain a single soil health value**

Application of simulation models of the soil-water-atmosphere-plant system can integrate the values of the parameters
mentioned above as they function as input data for the model, producing a single, integrated value for biomass production.
Many operational models are available (e.g., Reynolds et al., 2018; SWAP by Kroes et al., 2017; SWAP-WOFOST by Hack-
ten Broeke et al., 2019; ICASA by White et al., 2013; APSIM by Holzworth et al., 2018; Ma et al., 2012 and others). These
models use rooting depth, weather data and when the required hydraulic conductivity and moisture retention data are not
available, these values can be estimated with pedotransfer functions using texture (as defined by the soil type), % organic
matter and bulk density as input data, the soil health parameters identified above (Bouma, 1989; Van Looy et al., 2017). So





rather than have sets of separate parameters for soil health, an integrated expression is obtained by the model that directly
addresses a key soil function, which is its contribution to the ecosystem service "biomass production". The term "contribution"
needs to be emphasized as "biomass production" is not determined by soils alone but by many other factors and, certainly, by
management. Applying modelling, an alternative procedure to define soil health was proposed by Bonfante et al. (2019) where
biomass production forms the starting point. Following the agronomic Yield Gap program (van Ittersum et al., 2013) yields
are calculated by simulation models of the soil-water-atmosphere-plant system: $Yp$ = potential production determined for a
representative crop considering radiation and temperature regimes in a given climate region, assuming that adequate water and
nutrients are available and pest and diseases do not occur. This is a science- based value that applies everywhere on earth and
yields unique, quantitative and reproducible data. $Yw$ is the water-limited yield, as $Yp$, but expressing the effect of the actual
soil water regime under local conditions, and $Ya$ is the actual yield. The yield gap is $Yw-Ya$. These parameters of the Yield-
gap program can be applied to define soil health and soil quality parameters to be discussed in the next section but need to be
modified to express the specific impact of the soil.
Simulation modelling offers the possibility to express soil functioning, as mentioned in the definition of soil health, by an
interdisciplinary modelling effort with input by agronomists, hydrologists and climatologists, each providing basic data for the
models. This yields one number, based on an interdisciplinary analysis, which is preferable to a series of separate numbers for
soil parameters only as in the US systems. The soil science discipline presents the parameters, mentioned above, to the
interdisciplinary research team in the context of a well defined soil type that defines moisture regimes and rooting patterns.
This way, the soil type functions as a "carrier of information" or a "class-pedotransfer function" (Bouma, 1989).
Moreover, and more importantly, modelling is the only option to explore possible future effects of climate change on soil
health and soil quality, as will be demonstrated below. Procedures to define single soil health and soil quality parameters will
be presented in the materials and methods section of the paper.

## 1.5 Targeting soil health and soil quality towards the SDGs and the Green Deal by focusing on ecosystem services

The discussion of soil health and soil quality so far focused on the soil and the way it functions, mentioning goals such as
"biological productivity and environmental quality" (soil quality) and "vital soils that sustain plants, animals and humans"
(soil health). As mentioned in the introduction, since 2015, 193 countries have made a United Nations-initiated commitment
to reach seventeen Sustainable Development Goals (SDGs). The European Union launched its Green Deal in 2019. The soil
quality and soil health concepts are no meaningful goals by themselves and can obtain societal significance when linked to the
SDGs and the Green Deal. But there is no direct link, if only because soil management plays a key role in achieving the SDGs
and the goals of the Green Deal. The challenge for soil science is to explore ways in which healthy soils can contribute to
improving a number of key ecosystem services, that, in turn, contribute to the SDGs (e.g., Bouma, 2014; Keesstra, 2016). This
is important because SDGs and goals of the Green Deal are not only determined by ecosystem services but also by e.g., socio-
economic and political factors that are beyond control by sciences studying crop growth. Attention for the SDGs and the Green
deal implies attention for not only biomass production (SDG 2: zero hunger) but also for other ecosystem services that relate





directly to environmental quality, such as the quality of ground and surface water (SDG6: clean water and sanitation), carbon
sequestration and reduction of greenhouse-gas emissions for climate mitigation (SDG 13: climate action) and biodiversity
preservation (SDG 15: life on land). That is why the following definitions of soil health and soil quality are proposed:
• *Soil health is the actual capacity of a particular soil to function, contributing to ecosystem services*
• *Soil quality is the inherent capacity of a particular soil to function, contributing to ecosystem services.*
Both general definitions focus on soil contributions to ecosystem services that, in turn, contribute at this point in time to the
realization of the United Nations Sustainable Development Goals and the goals of the EU- Green Deal.
The four ecosystem services, mentioned above, have a different character. Biomass production (SDG 2) is governed by climatic
conditions and soil water regimes as characterized by modelling that yields quantitative and reproducible results for Yp and
Yw. Management plays a key role in determining Ya, and the other ecosystem services and is characteristically different for
different soil types. Clean water (SDG 6) can e.g., be obtained by precision fertilization, minimizing nutrient leaching to the
groundwater, while combatting erosion can minimize surface water pollution. But there are, in contrast to Yp or Yw values
for biomass production, no theoretical reference values for this ecosystem service, only threshold values of water quality by
environmental laws and regulations. This also applies to carbon sequestration and reduction of greenhouse gas emissions (SDG
13) and to life on land (SDG 15) for which as yet no environmental laws have been introduced. Different soils in different
climate zones will offer different challenges and opportunities to be met by appropriate management.
**2 Materials and methods**
**2.1 The Soil–Water–Atmosphere–Plant (SWAP) model**
The Soil–Water–Atmosphere–Plant (SWAP) model (Kroes et al., 2017) was applied to solve the soil water balance during
maize cultivation under estimated climate change and soil % SOM scenarios of Ap horizons. SWAP is an integrated physically-
based simulation model of water, transport in the saturated–unsaturated zone in relation to crop growth It assumes
unidimensional vertical flow processes and calculates the soil water flow through the Richards equation. Soil water retention
θ(h) and hydraulic conductivity k(θ) relationships as proposed by van Genuchten (1980) were applied. The unit gradient was
set as the condition at the bottom boundary. The upper boundary conditions of SWAP in agricultural crops are generally
described by the potential evapotranspiration ETp, irrigation and daily precipitation. Potential evapotranspiration was then
partitioned into potential evaporation and potential transpiration according to the LAI evolution, following the approach of
Ritchie (1972). The water uptake and actual transpiration were modeled according to Feddes et al., (1978), where the actual
transpiration declines from its potential value through the parameter α, varying between 0 and 1 according to the soil water
potential.

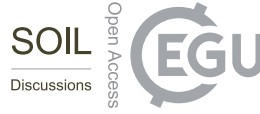

## 2.2 Soil Health and Soil Quality indicators

Application of the soil-water-atmosphere-plant simulation model and the yield-gap parameters results in four characteristics:

(i) a measure for actual soil health of a given soil type in a given climate zone at a given time by the SH index:

$$SH = (Yw - phenoform\ /Yw - ref) \cdot 100 \qquad [1]$$

where Yw-phenoform expresses Yw for a given phenoform and Yw-ref represents the undisturbed soil phenoform. This index expresses the effect of the soil on the measured yield Ya, a value that is affected by many other factors than the soil;

(ii) a measure for intrinsic soil quality (SQp) for a given soil type in a given climate zone, reflecting a characteristic range of soil health values obtained at different locations (SHL)as a function of different types of management (SHM) applied to that particular soil type, resulting in different phenoforms (p).

$$SQp = f(SHL, SHM) \qquad [2]$$

An example for three Italian soils will be shown later in figure 2.

(iii) a measure for intrinsic soil quality for all soils occurring in a given region in the same climate zone (SQr):

$$SQr = (Yw\ /Yp) \cdot 100 \qquad [3]$$

allowing comparisons among different soils in the region, with an option to again exprss effects of different phenoforms, and:

(iv) a measure for intrinsic soil quality allowing comparisons among all soils in the world in different climate zones (SQw):

$$SQw = (Yw\ /Ymax) \cdot 100 \qquad [4]$$

Values (ii) through (iv) can also be derived for different climate scenarios up to the year 2100, as reported by the Intergovernmental Panel on Climate Change (IPCC, 2014).

## 2.3 An Italian case study

Six prominent Italian soil series were analysed to illustrate the proposed method to define soil health and soil quality. Because of space constraints results of three soils will be discussed in this paper. The modeling process and the background of the IPCC scenarios have been presented elsewhere (Bonfante et al., 2019, 2020; Bonfante and Bouma, 2015) and will be summarized below.

The maize was simulated from May (emergence) to the end of August (harvest) with a peak of leaf area index (LAI) of 5.8 m$^2$ m$^{-2}$. Finally, the above ground biomass (AGB) to determine the yield values (Yw) was estimated using the normalized water productivity concept (WP; 33 g m$^{-2}$ for maize; Steduto et al., 2012).

The simulation runs were performed for six selected soils using a future climate scenario of a site of southern Italy (Destra Sele plain) where half of the analysed soils occur. The future climate scenarios were obtained by using the high resolution regional climate model (RCM) COSMO-CLM (Rockel et al., 2008), with a configuration employing a spatial resolution of 0.0715°(about 8 km), which was optimized over the Italian area. The validations performed showed that model data agree





closely with different regional high-resolution observational datasets, in terms of both average temperature and precipitation
(Bucchignani et al., 2015) and in terms of extreme events (Zollo et al., 2015).
The severe Representative Concentration Pathway (RCP) 8.5 scenario was applied, based on the IPCC modelling approach to
generate greenhouse gas concentrations (Meinshausen et al., 2011).
The results were performed on reference climate RC (1971–2005) and RCP 8.5, the latter divided into three different time
periods (2010–2040, 2040–2070 and 2070–2100). Daily reference evapotranspiration ($ET_0$) was evaluated according to the
Hargreaves and Samani (1985) equation.
Under the RCP 8.5 scenario, the temperature in Destra Sele is expected to increase approximately two degrees Celsius,
respectively, every 30 years to 2100, starting from the RC. The differences in temperature between RC and the period 2070–
2100 showed an average increase in the minimum and maximum temperatures of about 6.2°C (for both min and max over the
year). The projected increase in temperatures produces an increase in the expected $ET_0$. In particular, during the maize growing
season, an average increase of $ET_0$ of about 18% is expected until 2100 (Bonfante et al., 2020).
Simulations were run considering an undisturbed soil (the reference) and three phenoforms: two expressing degradation
phenomena (erosion and compacted plowpan) and one considering an increase of % OM in the first soil horizon (Ap), as a
possible result of combatting a low % OM due to soil degradation.
In particular:
(i)   The compacted plowlayer was applied at 30 cm depth (10 cm of thickness) with the following physical characteristics:

239       $\theta s$=0.30 cm³cm⁻³, n=1.12, α=0.004 and $k_0$=2 cm day⁻¹, following the notation of van Genuchten (1980). Roots were

240       restricted to the upper 30 cm of the soil.

(ii)   Erosion was simulated for the Ap horizon, reducing the upper soil layer to 20 cm. The maximum rooting depth was

242       assumed to be 60 cm (A+B horizons) with a higher root density in the Ap horizon.

(iii)   The effect of the increase of SOM to 4% on the first soil horizon (Ap) on hydraulic properties was realized applying

244       the procedure developed and reported in Bonfante et al. (2020) on hydraulic properties measured in the lab.

### 2.3.1 Soil characteristics

The Italian soils are located in a plain in an alluvial environments, two in the Campania region (P5 and P6) and P4 in the
Lombardy Region. The physical properties of the three selected soils are presented in Table 1. Soil texture range from sandy
loam to loamy sand and organic matter contents in Ap horizons are relatively low, ranging from 1.4 to 2.6%, justifying runs
for hypothetical contents of 4%. Based on field observations, the rooting depth of maize was estimated to be 80 cm, implying
that not the only Ap horizon but also subsoil horizons contribute to the water supply to maize.
The soil hydraulic properties applied in the simulation runs, water retention, θ(h), and hydraulic conductivity, k(θ), curves
were measured in the laboratory. Undisturbed soil samples (volume ≈ 750 ml) were collected from all of the recognized
horizons of the six soil profiles. Samples were slowly saturated from the bottom and the saturated hydraulic conductivity
measured by a falling head permeameter (Reynolds et al., 2002). Then, both couples of θ-h and k-θ data were obtained by





means of the evaporation method (Arya, 2002) consisting of an automatically recorded of the pressure head at three different
depths and the weight of the sample during a 1-dimensional transient upward flow. From these information, i) the water
retention data θ-h were obtained applying an iterative method (Basile et al., 2012) and ii) the unsaturated hydraulic conductivity
data were obtained by applying the instantaneous profile method, requiring the spatio-temporal distribution of θ and h, namely
θ(z,t) and h(z,t), being z and t the depth and time, respectively (Basile et al., 2006). Additional points of the dry branch of the
water retention curve were determined using a dewpoint potentiometer (WP4-T, Decagon Devices, Washington, USA).
The parameters of the van Genuchten-Mualem model for water retention and hydraulic conductivity functions were obtained
by fitting the experimental θ-h and k-θ data points (Van Genuchten, 1980).

## 3 Results

The emphasis in this paper will be on the application of the soil health and soil quality definitions presented above. Initially,
three adverse effects of management were considered: surface runoff caused by relatively low infiltration rates, erosion of 20
cm of topsoils (while soil classification remains the same), and formation of a plowpan at 30 cm depth (see Bonfante et al.,
2019). Results showed, however, that under prevailing current and future climate conditions surface runoff was negligible.
Results will therefore only be presented for phenoforms showing effects of erosion and the plowpan and for increaed %OM,
as mentioned above.

### 3.1 Water-limited yields (Yw)

Water-limited yields (Yw) for four climate periods and three phenoforms for each soil are shown in Figure 1a for soil P4,
Figure 1b for soil P5, and Figure 1c for soil P6. Yw values drop for all soils and their phenoforms in the period from the RC
to the 2070 -2100 climate scenario, particularly for climate scenarios beyond 2040, but due to relatively high standard
deviations, not all differences are significant. However, each soil shows significant drops of Yw for the erosion and plowpan
phenoforms, again particularly beyond 2040, when comparing values with Yw undisturbed. Soils P4 and P5 show rather
identical behavior but soil P6 has significantly higher values for Yw for the erosion and plowpan phenoforms beyond 2040.
An increase of % OM has minimal effect as explained by Bonfante et al. (2020) when considering hydraulic conductivity and
moisture retention data.

### 3.2 Soil health values for different climate periods

The SH index applies to soil health parametera measurements for a given soil at a given time, defining actual conditions with
reference to the particular production potential of the soil type that is present as expressed by Yw calculated with optimal soil
parameters as discussed above. Yw-phenoform conveys conditions, expressed by the three soil parameters observed at the site.
When Yw-phenoform is equal to Yw, the soil health value will be 100, but this is highly unprobable. Lower values indicate
room for improvement but offer no information as to factors that lead to these low values (see next section). Calculated SH





indexes for three Italian soil series in four climate periods are reported in Table 2. In this study, four soil conditions were
simulated that are common in the field, considering four climate periods: a non-degraded soil characterized by optimal soil
parameters (producing Yw-ref), and two Yw-phenoform values: erosion of topsoil, formation of a plowpan, and an increase
to 4% OM. As actual conditions are discussed here, the current climate of 2010-2040 should be considered. Erosion reduces
SH to appr. 88, while the plowpan has much stronger effect with significantly different values of 55 (soil P4), 66 (P5), and 75
(P6). Increasing % OM does not deviate from the value of 100, which corresponds with data reported in Figures 1, 2, and 3.
To determine the health index at a given time and place in a given soil, the three soil parameters discussed above are measured
and the model is used to calculate a (Yw-phenoform) value that is next compared with the Yw-ref value calculated with optimal
soil parameneter values for that particular soil. Management practices should be documented that have resulted in the Yw-
phenoform being considered.

**3.3 Soil quality (SQp) in terms of characteristic ranges of soil health values**

The SH index, mentioned in the previous section, characterizes soil health at a given time and location, as measured in a
particular soil type. A gap may become obvious between Yw-phenoform and Yw-ref but it is not clear what can be done to
close the gap. Soil health values for a given soil series can also be obtained at different locations in the same climate zone
where different forms of management have resulted in different phenoforms representing a characteristic range of values that
can be seen as a measure for inherent soil quality (SQp). Figure 2 shows a range of values obtained for a given soil type
assuming, in this case, the occurrence of only three phenoforms. This only illustrates a principle and many observations in the
field can and should extend the number of points for Yw-phenoform. This range offers a point of reference for each
observation, as discussed in the previous section, and allows conclusions as to advisable management procedures associated
with the different phenoforms that, together, determine the observed ranges in Figure 2.
Figure 2 shows a decreasing sensitivity for soil degradation moving from soil P4 to soil P6. Soil health ratios change from 56
(P4), 66 (P5) to 78 (P6). The effects of climate change on the index are, again, strongest for soil P4. Figure 2 shows that not
only the ranges of the health index are significantly different for the three soils but also their resilience to climate change. A
particular soil health measurement in a given soil, as described in the previous section, can now be placed into the bar shown
in Figure 2 indicating possible room for improvement. As every measurement is combined with an assessment of soil use and
management that has resulted in the particular phenoform being observed, the system allows the generation of useful
management information for the land user.

**3.4 Comparing different soils in a given region (SQr).**

So far, particular soil types have been considered. The analysis can be extended to all soils in a given region and climate zone
and this comparison of different soils can be valuable for regional land use planning. This requires the definition of Yp for the
area that is used for the simulations. For the Italian soils being considered Yp=18 tons ha$^{-1}$ and this value is maintained for all
climate scenarios considered, implicitly assuming that other factors affecting biomass production will not change. Table 3





shows significant differences among the soils providing a valuable quantitative assessment. Differences are maintained when different climate periods are considered. Soil P4 scores again the lowest values, with soil P5 intermediate and soil P6 with the highest values but even this soil has a low score of 50 for the last climate period when a plowpan is present.

**3.4 How to assess soil quality in a global context? (SQw).**

Questions about potential food production in future, considering the effects of climate change require a mechanism to compare different soils in the world in their capacity to produce biomass. Assuming a maximum production to be achieved in the world (Ymax) considering theoretical photosynthesis under particular climate conditions, values of Yp and Yw can be expressed as a function of Ymax. Use of Yw will produce the most realistic values in view of the limited water availability in many areas of the world. Areas with relatively high values have a higher potential than areas with low values and this analysis can be helpful input from soil science contributing to global food production scenarios. Based on current evaluations, a Ymax of 20 tons ha$^{-1}$ is used here as a reference and this results in SQw values that can also be expressed for various phenoforms, showing effects of different forms of degradation Table 4. As in Table 3, differences between the three soils are significant. How these values are to be judged will depend on comparable values to be assembled for other areas of the world.

**4 Discussion**

The Soil Health concept, as defined in the literature and as modified in this study, is inadequate to allow a comparison of the capacity of different soils to function. Two soils may be healthy in their own way, but a healthy clay soil has a significantly different "capacity to function" as compared with a healthy sandy soil. Still, the soil health concept is suitable to express the actual condition of a given soil by comparing Yw-phenoform with Yw-ref as discussed in this paper, producing a soil health index SH. The advantage of this procedure is its basis in a quantitative and reproducible scientific analysis of the plant production process as a function of soil moisture regimes, made possible by applying soil-water-atmosphere-plant simulation models. Yw-ref and Yw-phenoform reflect the impact of soil conditions on Ya, the measured yield, as water and nutrients are assumed to be optimal and pests and diseases do not occur. Observing the difference between Ya on the one hand and Yw-phenoform and Yw-ref on the other can result in fruitful interaction between soil scientists and agronomists applying a common language as an effective means of communication.

When applied to three Italian soils, defined by soil classification in terms of three genoforms, a range of values is obtained not only for an undisturbed soil but also for soils affected by poor forms of soil management resulting in erosion and compaction (two "phenoforms"), and a third phenoform following "good" management increasing % OM. All of these phenoforms still maintain their genoform classification (Bouma, 1989; Rossiter and Bouma, 2018). In this study effects of only three hypothetical phenoforms were explored. In future, field work is required to distinguish a number of characteristic phenoforms for every genoform, as a function of current and past soil management. Existing soil maps can be used to identify sampling spots (e.g., Pulleman et al., 2000; Sonneveld et al., 2002).





Again, the different soils show significantly different behavior and the ranges for each soil series, reflecting the effects of
management, are different. This range represents an inherent property of the soil series being considered and it is a de facto
measure for soil quality (SQp) as expressed in Figure 2 and adds an important element to soil survey interpretations that are
now empirical and qualitative in terms of "general suitabilities or limitations for various forms of land use". This requires that
properties of phenoforms are explained in terms of management practices. In this context, Pullemnan et al. (2000) and
Sonneveld et al. (2002) successfully correlated present and past management with % organic matter in topsoil.
When considering the use of soils in a given region, the SQr, as defined above, is helpful to compare the production potential
of different soils in that particular  region
Finally, analyses on the world level can be made by considering the SQw index, expressing local Yw-ref values (if so desired
subdivided in terms of relevant phenoform values) versus a global upper limit. This could be a valuable absolute procedure to
compare soils on world level which may be relevant when considering future world food supply scenarios, allowing a focus
on potentially favorable locations. This provides an added value to the "yield-gap" program that focuses on reducing the gap
(van Ittersum et al., 2013).
However, as stated in the introduction, soil health and soil quality are no objectives in themselves. Achieving the UN
Sustainable Development Goals and the goals of the EU Green Deal require that soils provide effective contributions to various
ecosystem services that, in turn, contribute to SDGs and the Green Deal. Soils function in an interdisciplinary context and the
implicit hypothesis of soil health assumes that healthy soils will make better contributions to ecosystem services than unhealthy
ones and soils with low quality in a regional and world context. But a healthy soil can still make a poor contribution to
ecosystem services when poorly managed, illustrating the overriding importance of the management factor.
Application of soil-water-atmosphere-plant models is focused on the ecosystem service: "biomass or primary production".
However, at the same time, other services have to be provided as well as discussed earlier: water quality protection, reduction
of greenhouse gas emissions, carbon capture and biodiversity preservation. Here, applying appropriate management is crucial
and, in contrast to the calculations of biomass production, there is no underlying basic theory to identify options. That is why
defining a characteristic range of soil health values for any given soil types a measure for inherent soil quality (SQp) is
important to link the land user with experiences obtained elsewhere on similar soils in the same climate zone.

**5 Conclusions**


1. Focusing on actual conditions when defining soil health and on inherent conditions when defining soil quality allows a
meaningful distinction between the two concepts that are both needed.
2. Introduction of the terminology of the agronomic "yield gap" program, allows quantitative and reproducible expressions
for the soil health and soil quality.concepts. The distinction of Yw-ref and Yw-phenoform allows independant estimates
of soil contributions to Ya, the actual yield (=ecosystem service: biomass production) that is determined by many other
factors disciplines than soil. Applying the "yield-gap" terminology will facilitate interaction with agronomists.



3. The soil health and soil quality concepts have societal relevance as they contribute to defining ecosystem services that, in turn, contribute to the UN-SDGs and the EU Green Deal.

4. Soil types were effective "carriers of information" (class-pedotransfer functions) showing distinctly different values for the soils being considered.

5. Effects of climate change on Yw were significant for the Italian soils being considered with projected reductions in productivity, also for non-degraded soils including soils with higher organic matter contents that may not allow economically viable forms of agriculture by the end of the 21thcentury if irrigation is not feasible.

6. Even healthy soils can fail in making significant contributions to ecosystem services when poor management is applied. Soil use and management play a key role when interpreting soil health and soil quality indexes by providing advise as to how to increase indexes. The effects of soil use and management on a given type of soil (genoform) can be expressed by defining phenoforms of particular genoforms. This will require new fieldwork guided by existing soil maps.

7. Effects of climate change on Yw were significant for the Italian soils being considered with projected reductions in productivity, also for non-degraded soils including soils with higher organic matter contents, that may not allow economically viable forms of agriculture by the end of the 21thcentury if irrigation is not feasible.

**6 Acknowledgements.**

Acknowledge Mrs. N. Orefice and Dr. R. De Mascellis for soil hydraulic property measurements and Dr. Eugenia Monaco for the support in the analysis of climate scenarios. Climate data from the "Regional Models and Geo-Hydrogeological Impacts Division" of the Centro Euro-Mediterraneo sui Cambiamenti Climatici (CMCC), Capua (CE) – Italy, were applied in this study, with support by Dr. Paola Mercogliano and dr. Edoardo Bucchignani. Finally, a special thanks to Dr. Guido Rianna for climate data analysis support.

LANDSUPPORT

**Funding:** This research was funded by EC H2020 LANDSUPPORT project, grant number 774234

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




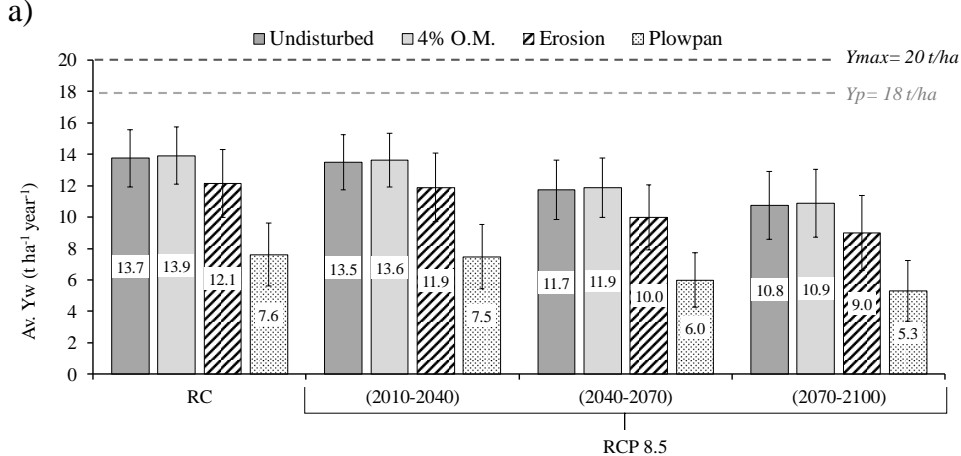

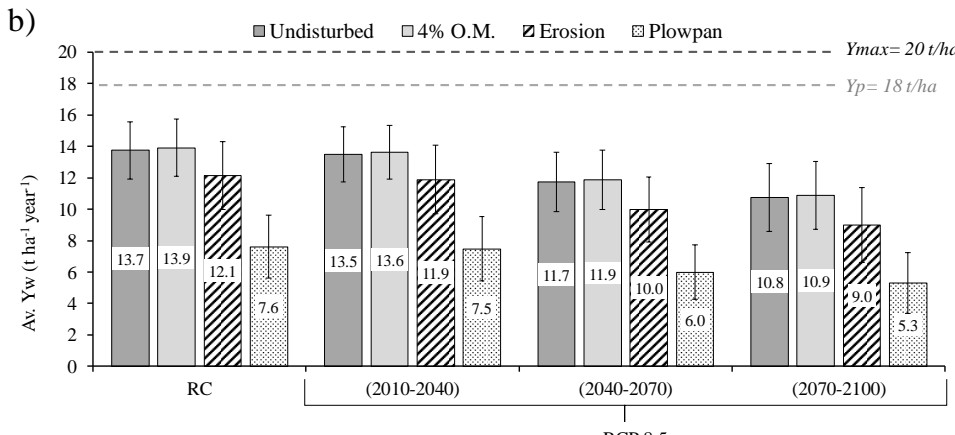

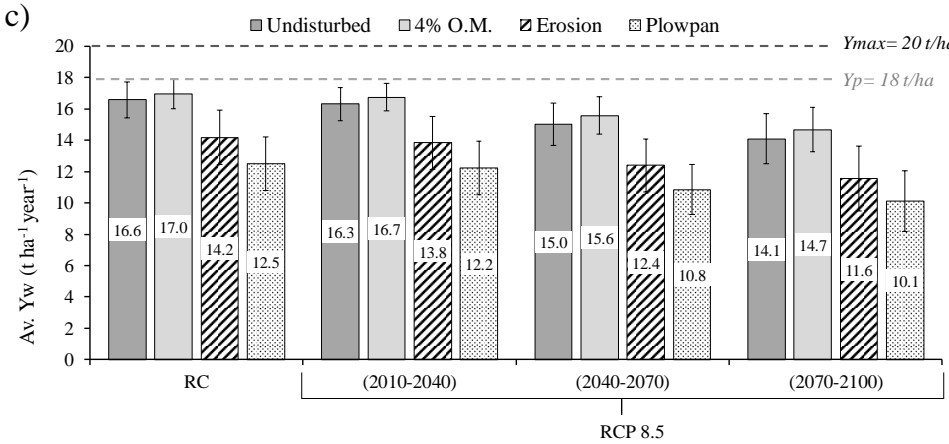


**Figure 1: The average Yw of four soil phenoforms of three soils (a) P4, (b) P5, and (c) P6 under reference (RC) and future climate**
**scenario (RCP 8.5). Yp is the local current potential production and Ymax is the maximum potential production under no stressed**
**field conditions (water, nutrient and pests disease).**



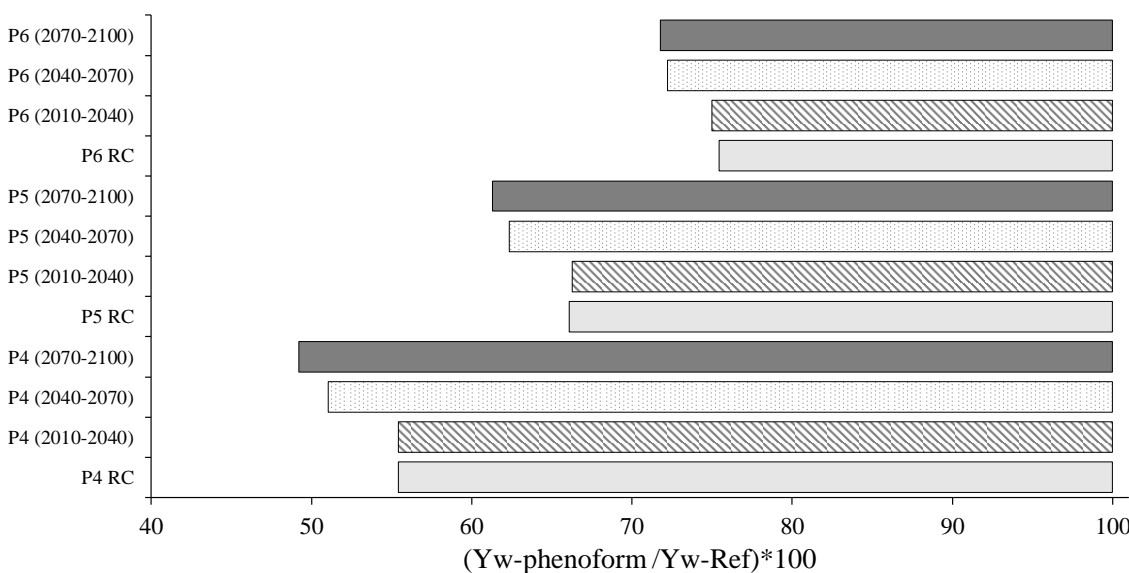


**Figure 2. Range of soil health indexes – SH=(Yw-phenoform/Yw-ref) x 100 - for the three soils demonstrating differences among soils and projected effects of climate change. This range characterizes the inherent soil quality SQp for these particular soil types.**





Tab. 1 Physical characteristics and classifications of the three Italian soils being studied (from Bonfante et al., 2020).

| Soil | | | Hor. | Thick. (cm) | Clay | Silt | Sand | S.O.M. |
|---|---|---|---|---|---|---|---|---|
| ID | Series | Classification | | | | % | | |
| P4 | Sordio[+] | Ultic Haplustalf, coarse loamy, mixed, mesic (Sandy Loam) | Ap1 | 0-18 | 17.9 | 32.6 | 49.5 | 1.4 |
| | | | Ap2 | 18-30 | 17.7 | 33.2 | 49.1 | 1.4 |
| | | | Bt1 | 30-56 | 21.8 | 31.4 | 46.8 | 0.4 |
| | | | Bt2 | 56-83 | 13.4 | 12.1 | 74.5 | 0.2 |
| | | | BC | 83+ | 10.0 | 6.3 | 83.7 | 0.1 |
| P5 | Masseria Manfredi[++] | Typic Ustivitrands, sandy, mixed, thermic (Sandy Loam) | Ap1 | 0-10 | 10.5 | 38.5 | 51.0 | 2.6 |
| | | | Ap2 | 10-40 | 5.9 | 43.6 | 50.5 | 2.6 |
| | | | Bw | 40-80 | 3.9 | 31.1 | 65.0 | - |
| | | | BC | 80-110 | 11.6 | 15.4 | 73.0 | - |
| | | | C | 110+ | 4.6 | 9.4 | 86.0 | - |
| P6 | Masseria Battaglia[++] | Vitrandic Haplustept, sandy, mixed (Loamy Sand) | Ap1 | 0-20 | 4.1 | 18.6 | 77.3 | 1.7 |
| | | | Ap2 | 20-53 | 6.1 | 18.4 | 75.5 | 1.6 |
| | | | Bw1 | 53-61 | 1.4 | 12.4 | 86.2 | 0.9 |
| | | | Bw2 | 61-106 | 2.2 | 8.7 | 89.1 | 0.9 |
| | | | C | 106+ | 1.0 | 24.6 | 74.4 | 0.2 |

+ *Soil series The soil map of Lodi plain (1:37.500) (ERSAL, 2000)*

++ *Closed to soil series of "The soil map of province of Naples" (1:75.000) (Di Gennaro and Terribile, 1999)*







Tab.2. Table 2. Soil health indexes - SH (( Yw-phenoform/Yw-ref) x 100), defining actual conditions, for three selected soils being studied for four cimate periods as indicated. Values are reported for the non-degraded soil and for hypothetical phenoforms representing , erosion of 20 cm of topsoil without a change of soil classification (Yw-erosion) and occurrence of a plowpan at 30 cm depth (Yw-plowpan) Indexes are also included for hypothetically increased % organic matter to levels of 4% ,(Yw-4% O.M.).

| Soil | Climate scenario | | Yw-erosion | Yw-Plowpan | Yw- 4% O.M. |
|------|------------------|---|------------|------------|-------------|
| P4 | RC | (1971-2005) | 88.4( ± 2.0) | 55.4 ( ± 1.9) | 101.1 ( ± 1.8) |
| | RCP 8.5 | (2010-2040) | 88.0 ( ± 1.9) | 55.4 ( ± 1.9) | 101.0 ( ± 1.7) |
| | | (2040-2070) | 85.1 ( ± 2.0) | 51.0 ( ± 1.8) | 101.1 ( ± 1.9) |
| | | (2070-2100) | 83.7 ( ± 2.3) | 49.2 ( ± 2.0) | 101.2 ( ± 2.2) |
| P5 | RC | (1971-2005) | 88.9 ( ± 1.7) | 66.1 ( ± 1.8) | 100.7 ( ± 1.6) |
| | RCP 8.5 | (2010-2040) | 88.9 ( ± 1.6) | 66.3 ( ± 1.7) | 100.7 ( ± 1.5) |
| | | (2040-2070) | 87.0 ( ± 1.7) | 62.3 ( ± 1.7) | 100.8 ( ± 1.7) |
| | | (2070-2100) | 86.7 ( ± 2.0) | 61.3 ( ± 2.0) | 100.8 ( ± 1.9) |
| P6 | RC | (1971-2005) | 85.5 ( ± 1.4) | 75.4 ( ± 1.4) | 102.4 ( ± 1.0) |
| | RCP 8.5 | (2010-2040) | 84.9 ( ± 1.4) | 75.0 ( ± 1.4) | 102.7 ( ± 1.0) |
| | | (2040-2070) | 82.5 ( ± 1.5) | 72.2 ( ± 1.5) | 103.7 ( ± 1.3) |
| | | (2070-2100) | 82.1 ( ± 1.8) | 71.8 ( ± 1.8) | 104.2 ( ± 1.5) |





Table 3. SQr index ((Yw/Yp) x100) for the three selected soils and the four climate periods.
Yp is assumed to be 18 tons ha$^{-1}$.

| Soil | Climate scenario | | Soil phenoform | | | |
|------|------------------|--|-------------|---------|----------|---------|
| | | | Undisturbed | 4% O.M. | Erosion | Plowpan |
| | | | | (Yw/Yp) x 100 | | |
| P4 | RC | (1971-2005) | 76.3 (± 1.8) | 77.2 (± 1.8) | 67.4 (± 2.1) | 42.3 (± 2.0) |
| | RCP 8.5 | (2010-2040) | 74.9 (± 1.7) | 75.7 (± 1.7) | 66.0 (± 2.1) | 41.5 (± 2.0) |
| | | (2040-2070) | 65.2 (± 1.8) | 65.9 (± 1.8) | 55.4 (± 2.0) | 33.2 (± 1.7) |
| | | (2070-2100) | 59.7 (± 2.1) | 60.4 (± 2.1) | 50.0 (± 2.3) | 29.4 (± 1.9) |
| P5 | RC | (1971-2005) | 83.1 (± 1.6) | 83.6 (± 1.5) | 73.8 (± 1.8) | 54.9 (± 1.9) |
| | RCP 8.5 | (2010-2040) | 81.4 (± 1.4) | 82.0 (± 1.4) | 72.4 (± 1.8) | 53.9 (± 1.9) |
| | | (2040-2070) | 72.9 (± 1.6) | 73.5 (± 1.6) | 63.5 (± 1.7) | 45.4 (± 1.7) |
| | | (2070-2100) | 67.8 (± 1.9) | 68.4 (± 1.9) | 58.8 (± 2.1) | 41.5 (± 2.0) |
| P6 | RC | (1971-2005) | 92.0 (± 1.1) | 94.2 (± 0.9) | 78.7 (± 1.7) | 69.4 (± 1.7) |
| | RCP 8.5 | (2010-2040) | 90.6 (± 1.0) | 93.0 (± 0.8) | 76.9 (± 1.6) | 67.9 (± 1.6) |
| | | (2040-2070) | 83.4 (± 1.3) | 86.5 (± 1.2) | 68.8 (± 1.6) | 60.2 (± 1.5) |
| | | (2070-2100) | 78.2 (± 1.5) | 81.5 (± 1.4) | 64.2 (± 2.0) | 56.1 (± 1.9) |





Table 4. SQw index ((Yw/Ymax)x100) for the three selected soils and the four climate periods. Ymax is assumed to be 20 tons ha$^{-1}$.

| Soil | Climate scenario | | Soil phenoform | | | |
|------|------|------|------|------|------|------|
| | | | Undisturbed | 4% O.M. | Erosion | Plowpan |
| | | | (Yw/Ymax) x 100 | | | |
| P4 | RC | (1971-2005) | 68.7 ( ± 1.8) | 69.4 ( ± 1.8) | 60.7 ( ± 2.1) | 38.0 ( ± 2.0) |
| | RCP 8.5 | (2010-2040) | 67.4 ( ± 1.7) | 68.1 ( ± 1.7) | 59.4 ( ± 2.1) | 37.4 ( ± 2.0) |
| | | (2040-2070) | 58.6 ( ± 1.8) | 59.3 ( ± 1.8) | 49.9 ( ± 2.0) | 29.9 ( ± 1.7) |
| | | (2070-2100) | 53.7 ( ± 2.1) | 54.4 ( ± 2.1) | 45.0 ( ± 2.3) | 26.4 ( ± 1.9) |
| P5 | RC | (1971-2005) | 74.8 ( ± 1.6) | 75.3 ( ± 1.5) | 66.4 ( ± 1.8) | 49.4 ( ± 1.9) |
| | RCP 8.5 | (2010-2040) | 73.3 ( ± 1.4) | 73.8 ( ± 1.4) | 65.1 ( ± 1.8) | 48.5 ( ± 1.9) |
| | | (2040-2070) | 65.6 ( ± 1.6) | 66.2 ( ± 1.6) | 57.1 ( ± 1.7) | 40.9 ( ± 1.7) |
| | | (2070-2100) | 61.0 ( ± 1.9) | 61.5 ( ± 1.9) | 52.9 ( ± 2.1) | 37.4 ( ± 2.0) |
| P6 | RC | (1971-2005) | 82.8 ( ± 1.1) | 84.8 ( ± 0.9) | 70.9 ( ± 1.7) | 62.5 ( ± 1.7) |
| | RCP 8.5 | (2010-2040) | 81.5 ( ± 1.0) | 83.7 ( ± 0.8) | 69.2 ( ± 1.6) | 61.1 ( ± 1.6) |
| | | (2040-2070) | 75.0 ( ± 1.3) | 77.8 ( ± 1.2) | 61.9 ( ± 1.6) | 54.2 ( ± 1.5) |
| | | (2070-2100) | 70.4 ( ± 1.5) | 73.3 ( ± 1.4) | 57.8 ( ± 2.0) | 50.5 ( ± 1.9) |
