# Peer review of "Targeting the soil quality and soil health concepts when aiming for the United Nations Sustainable Development Goals and the EU Green Deal"

_SOIL, 2020_

## Referee Comment (RC1) · Anonymous Referee #1 · 23 Jun 2020

The topic of the paper is very actual, and the definition of soil health and quality are still strongly debated. The authors explained their ideas about these definitions and proposed an innovative approach to calculate soil health and quality indices by using simulation models. The idea to use yield-gap concept to quantify the effect of soil health/quality on one of the most important soil functions, which is "biomass production"" is a good idea, in my opinion. Such approach could also allow to quantify the economical return of soil health and quality, in the future. For the actual topic and for the innovative approach to quantify the soil quality/health indices, the paper is worthy of

publication. There are only few sentences to clarify and to improve before publication, following the specific comments: 19: dot after "biomass production" 91: reported some references for measurement of SOM by proximal sensors, e.g. by Vis-NIR or others. 112-113: It's not clear as quality concept added values to health concept. Does soil quality take in account the texture? Soil health is calculated on three different soil texture classes, therefore it coarsely takes in account the texture. Please, explain better this part. 114-115: Since "Phenoforms" is a concept reported several times in the paper and in the eq. [1], probably you should better clarify if it is only a concept of "soil status (eroded, compacted etc.)" or if it can be also quantified with a number. It's not very clear for the readers. 181: A dot before "It assumes. . .." 194, eq[1]: it's not clear. From the equation, lower values seem to be calculated for healthy soils, because minimum yield gap between phenoform and reference. From the data reported in tab.2, higher values of SH seem to correspond to healthier soils. Please, clarify. In case of lower SH values corresponded to healthier soils, you should probably modify the equation (for example adding 1-), otherwise it is misleading. 204: "express" instead of "exprss" 280: "Parameters" instead "parametera" 355: dot at the end 359-360: Since most of your work is focused on the effects of soil on primary production function, you should introduce the economical importance of soil health and soil quality. Moreover, your work is based on soil series, but you should briefly introduce the concept of the "short range spatial variability" of the soils. Within a field, two or more soil types with very different soil health and quality could be there. A detailed soil map, also by the use of proximal and remote technologies, allow to characterize the soil spatial variability (and then SH and SQ) at high detail. The site-specific approach to preserve soil health and quality could be basic to save resources and yield in a climate change context. Please, try to briefly introduce this theme in your discussions.

---

## Referee Comment (RC2) · Anonymous Referee #2 · 29 Jul 2020

I commend the authors for their efforts to clarify and parameterize the concepts of soil quality and soil health. The framework presented in this manuscript to define these and a number of related terms is quite clear. In addition, the methods proposed to parameterize these terms are straightforward and considerably cleaner than many other methods that have been previously proposed, relying mainly on existing data, and requiring far fewer parameters and measurements than other soil quality and soil health "systems".

The manuscript is well-written, with only a few minor grammatical errors, noted below.

[Figure]

I am, however, surprised that soil P6, a loamy sand, has a higher Yw for the undisturbed condition (and all other conditions represented) than soils P4 and P5, which are sandy loams. Plant available water should be lower in a soil with loamy sand texture than one with sandy loam texture, which would suggest an overall lower yield for the loamy sand soil, all other things being equal. In my experience here in Minnesota, unless irrigated, loamy sand soils commonly have lower yields than sandy loam soils. Are these the correct data for soil P6?

Specific comments:

l. 377: remove period between quality and concepts

l. 378-379: I suggest changing the phrase: "...that is determined by many other factors disciplines than soil." to "...that is determined by many factors other than the soil (i.e., insect invasions, plant disease, other factors)."

l. 384-386. I suggest rewriting this point as: Effects of climate change for the Italian soils being considered showed such a significant and large reduction of Yw for all degraded and non-degraded scenarios considered that agriculture may not be economically viable by the end of the 21st century if irrigation is not feasible.

l. 386: 21st century, rather than 21thcentury

l. 391-393: Point 7 is identical to point 5. One of these two points should be deleted.

l. 398: dr. should be capitalized to Dr.

---

## Author Response (AR1)

**Answer to referee 1.**

We thank this referee for his positive comments. His remarks on the paper were quite relevant and helped us to improve the clarity and scope of the text.

1.      **Line 91:** We have added two references to proximal sensing. There are many but this one is particularly interesting as it relates to an East European joint study with US colleagues and considers soils in a landscape context.

2.      **Lines 112** and following: We have modified the text to show that our method to measure soil health is universal and applies to any soil at any time at any place. The referee is correct in stating that the Cornell protocol stratifies measurements by three texture classes and- important!- has separate frequency curves for the indicators for each class. We prefer a universal test for all soils resulting into comparable values rather than separate procedures for different texture classes. and we now state so clearly. We also feel that just distinguishing three texture classes does no justice to soil expertise that is available because there are major differences in soil behavior within each of these three very broad texture classes and that's why we present results for Italian soil series that provide much more info than just a texture class. We added a recent reference of Bouma (2020) that discusses use of soil data in models for those that want to read more about this. Next, we define soil quality in three ways for a given soil type, indicating the inherent character of the soil quality concept, showing a characteristic range of values for a given soil type, for different soils within regions and in the world at large, allowing comparisons among soils

3.      **Line 114 plus**: We have explained the phenoforms in some more detail now. To establish a value for SH and SQ based on so-called and undefined "representative profiles" of soil survey interpretations is not as good as distinguishing several phenoforms of that particular soil that reflect effects of management because soil management can have significant effects on soil behavior while the name for the soil ( the genoform) stays the same! Effects of management, as expressed by the phenoforms, strongly affect soil quality and cannot be ignored. The latter is important in our view and is now more clearly emphasized. This is illustrated in the presented figures of the Italian soil series. Of course, the examples are hypothetical and should in future be based on field research along the lines as presented by Sonneveld et al and Pulleman et al.

4.      **Line 194**: equation (1) is OK. Soils with lower SH have lower Yw-phenoform/ Yw-ref. ratios, because soil degradation lowers Yw. Healthy soils have higher values. This corresponds with the tables.

5.      A sentence is added mentioning the fact that soil production is directly related to economic aspects and a reference has been added. Also, we mention that once it has been shown that Yw is higher than Ya (the real yield) an analysis is needed to find out why this is the case and how management can be devised to overcome problems. A discussion of this is, however, beyond the scope of this paper as we mention in the revised text.

6.      Suggestions for changes in punctuation have been followed. Thanks.

**Answer to referee 2**

We appreciate the positive comments of the reviewer on our paper. Her/His specific comments were relevant and quite helpful and have been adopted in the revised manuscript. Her/His question as to why Yw of soil P6 (a loamy sand) is higher than the other two soils (sandy loams) can be explained by the following:

- "Plant available water", defined as the water content between two pressure heads, does not represent the volume of water that is available for plants in a given growing season, as explained in the quoted letter to the editor in the Eur. J. of Soil Science by Bouma (2018) and thoroughly in this manuscript. Here, we use a dynamic sink term to express water uptake by roots resulting in varying water extraction as a function of the water content. These processes are highly non-linear, strongly depending on the shapes of the moisture retention and hydraulic conductivity curves.
- Furthermore, soil hydraulic properties − strongly dependent by the soil structure and only partially by the soil texture − cannot be tightly related to the latter. In fact, even the widely applied pedo-transfer-functions (PTF) are not able to fully capture the intrinsic complexity of soil hydraulic properties, showing sometimes a weak correlation between measured and estimated parameters (r=0.3 − 0.5).
- For these reasons we performed the soil water balance simulations by applying measured hydraulic properties and not estimated ones. This approach is even more important for our case studies because some of our soils (e.g. P6) show distinct hydraulic properties (Basile et al., 2007) due to the presence of short-range ordered clay minerals (e.g. allophane). Moreover, these soils are rather difficult to disperse due to their high variable charges (Mizota & van Reeuwijk, 1989), making textural analysis rather uncertain.
- Finally, field experience in the area (Agro Nocerino-Sarnese plain, south of Vesuvius volcano) where this soil occurs indicates relatively accessible water, leading to high productions when, of course, other agronomic factors are optimal.

Specific comments:

l. 377: remove period between quality and concepts: *Done*

l. 378-379: I suggest changing the phrase: "...that is determined by many other factors disciplines than soil." to "...that is determined by many factors other than the soil (i.e., insect invasions, plant disease, other factors).".

*The sentence has been improved has requested by the reviewer*

**l. 384-386**. I suggest rewriting this point as: Effects of climate change for the Italian soils being considered showed such a significant and large reduction of Yw for all degraded and non-degraded scenarios considered that agriculture may not be economically viable by the end of the 21st century if irrigation is not feasible.

*The sentence has been improved has following:*

*Old sentence "Effects of climate change on Yw were significant for the Italian soils being considered with projected reductions in productivity, also for non-degraded soils including soils with higher organic matter contents that may not allow economically viable forms of agriculture by the end of the 21thcentury if irrigation is not feasible.".*

*New sentence (lines 403-405 new manuscript) "Effects of climate change for the Italian soils being considered showed such a significant and large reduction of Yw for all degraded and non-degraded scenarios, that agriculture may not be economically viable by the end of the 21st century if irrigation is not feasible."*

**l. 386:** 21st century, rather than 21thcentury: *Changed*

**l. 391-393:** Point 7 is identical to point 5. One of these two points should be deleted.: *The sentence at point 7 has been deleted.*

**l. 398:** dr. should be capitalized to Dr.

Bouma, J.: Comment on: B. Minasny & A.B. Mc Bratney. 2018. Limited effect of organic matter on soil available water capacity, Eur. J. Soil Sci., 69(1), 154–154, doi:10.1111/ejss.12509, 2018.

Basile, A., Coppola, A., De Mascellis, R., Mele, G., & Terribile, F. (2007). A comparative analysis of the pore system in volcanic soils by means of water-retention measurements and image analysis. In Ó. Arnalds, H. Óskarsson, F. Bartoli, P. Buurman, G. Stoops, & E. García-Rodeja (Eds.), Soils of volcanic regions in Europe (pp. 493–513). Berlin, Heidelberg: Springer.

Mizota, C., & van Reeuwijk, L. P. (1989). Clay mineralogy and chemistry of soils formed in volcanic material in diverse climatic regions. Soil Monograph, 2. Wageningen, NL: International Soil Reference and Information Centre (ISRIC).